# Molecular and Clinical Premises for the Combination Therapy Consisting of Radiochemotherapy and Immunotherapy in Non-Small Cell Lung Cancer Patients

**DOI:** 10.3390/cancers13061222

**Published:** 2021-03-11

**Authors:** Małgorzata Frąk, Paweł Krawczyk, Ewa Kalinka, Janusz Milanowski

**Affiliations:** 1Chair and Department of Pneumology, Oncology and Allergology, Medical University of Lublin, 20-954 Lublin, Poland; krapa@onet.pl (P.K.); janusz.milanowski@umlub.pl (J.M.); 2Department of Oncology, Polish Mother’s Memorial Hospital Research Institute in Lodz, 93-338 Lodz, Poland; ewakalinka@wp.pl

**Keywords:** non-small cell lung cancer, radiotherapy, immunotherapy, PD-1, PD-L1

## Abstract

**Simple Summary:**

Immunotherapy is one of the most effective systemic treatment methods for many types of cancers. Unfortunately, cancer cells developed a number of defense mechanisms e.g., the absence of NK cells, macrophages or T lymphocytes in the tumor stroma, lack of pro-inflammatory cytokines in the tumor microenvironment (IL-6, IL-2, IL-12, TNF-alpha), production of immunosuppressive compounds (TGF-beta, indoleamine dioxygenase or neutralization of immune cells through direct immune checkpoints interactions (CD80/CD86 with CTLA-4 and PD-L1 with PD-1) that eventually make treatment ineffective. In this way, non-immunogenic, “cold” tumors are formed. The paper presents those mechanisms in details and focuses on the radiochemotherapy technique which, by neoantigen production, abscopal effect and activation of interferon synthesis pathways (STING), affects the production of cytokines and chemokines and transforms “cold” tumors into highly immunogenic “hot”, inflammatory tumors, susceptible to immunotherapy. Results are based on clinical trials conducted to date, which showed high effectiveness of the combination therapy consisting of radiochemotherapy and immunotherapy in NSCLC patients.

**Abstract:**

Non-small cell lung cancer (NSCLC) is one of the most common malignancies around the world. Due to the advanced stage of the disease at the time of diagnosis, most patients require systemic treatment. Immunotherapy with immune checkpoints inhibitors is becoming the main treatment method for many cancers, including NSCLC. Numerous studies have shown greater efficacy of immunotherapy used monoclonal antibodies anti-PD-1 (pembrolizumab and nivolumab) or anti-PD-L1 (atezolizumab and durvalumab) compared to chemotherapy. Unfortunately, cancer cells can develop a number of mechanisms to escape from immune surveillance, including avoidance of cancer cells by the immune system (immune desert), production of immunosuppressive compounds (prostaglandins, IDO, TGF-beta), or direct immune checkpoints interactions. Therapy based on the use of radiochemotherapy with subsequent immunotherapy is becoming the main focus of research in the field of new NSCLC therapies. Radiation therapy stimulates the immune response multidirectionally, affects production of neoantigens and proinflammatory compounds, which transform non-immunogenic (“cold”) tumors into highly immunogenic (“hot”) tumors. As a result, the mechanisms of escape of cancer cells from immune surveillance break down and the effectiveness of immunotherapy increases significantly. The results of clinical trials in this area bring new hope and indicate greater effectiveness of such treatment in terms of prolongation of progression-free survival and overall survival.

## 1. Introduction

Non-small cell lung cancer accounts for around 80% of all lung cancer cases. The basic method of treatment is surgery procedure. Unfortunately, due to the lack of characteristic symptoms of the disease in an early stage, as well as the lack of effective screening tests, in most cases the disease is diagnosed in late stage of local progression or in advanced, metastatic stage. Only about 15–20% of patients are eligible for surgery, and over 80% of patients require a systemic method of treatment [1].

Immunotherapy, next to chemotherapy and molecularly targeted therapies, is becoming one of the main methods of systemic treatment for many types of cancers [1]. The first observations suggesting the effectiveness of immunotherapy in the treatment of cancer took place in the late nineteenth century, according to research conducted by the American surgeon William Coley. In these experiments, he observed that in cancer patients (including extremely malignant sarcomas) and the accompanying bacterial infection (e.g., erysipelas), spontaneous tumor remission occurred. In later studies, Coley obtained remission of various types of cancers due to deliberate infection of patients with erysipelas or by bacterial culture supernatants injections. This gave rise to non-specific active immunotherapy, used e.g., Bacillus Calmette-Guérin (BCG) vaccine for intravesical infusions in patients with surgically removed bladder cancer. Clinical studies have also been carried out using active antigen-dependent immunotherapy using tumor cell antigens or whole tumor cells. Unfortunately, the above forms of treatment did not bring the expected results.

The discovery of tumor escape mechanisms from immune surveillance-a theory presented in 1967 by Burnet and Thomas-opened a new era in the development of immunotherapies aimed at negative immune checkpoints, such as programmed death 1 (PD-1), programmed death ligand 1 (PD-L1), cytotoxic T lymphocyte antigen 4 (CTLA-4), lymphocyte-activation gene 3 (LAG-3), T-cell immunoglobulin mucin-3 (TIM-3). The presence of their expression on cancer cells, on immune cells infiltrating the tumor or lymph nodes, and on lymphocytes, causes the anergy of cytotoxic T lymphocytes (CTLs) and the lack of a specific anti-tumor response [2,3]. Blocking negative immunological checkpoints enhances the anti-tumor response by CTLs.

Under normal conditions, cancer cells are recognized by the immune system and then destroyed. Through evolution, cancer cells have developed mechanisms that compromise the immune response. This process is multi-stage and includes, first of all, avoiding the recognition of cancer cells, including reduction or lack of expression of major histocompatibility complex (MHC) class I molecules and their antigens, reduction or lack of expression of co-stimulatory molecules and expression of proteins associated with antigen processing and its transport to the cell membrane, and increase in expression of negative immune checkpoints [4]. As a result, tumor cells are not recognized by cytotoxic T cells, and no tumor infiltrates composed of CD-8 positive T cells are found in the tumor stroma. This phenomenon has been described as “immune desert” or “cold tumor” [5]. Low-immunogenic tumor is characterized by the absence of cells involved in non-specific (NK cells, macrophages, neutrophils) and specific (effector T lymphocytes, T regulatory cells, T helper cells) immune responses and a lack of chemotactic factors and pro-inflammatory cytokines in the tumor microenvironment (IL-6, IL-2, IL-12, TNF-alpha). A non-immunogenic tumor (“cold tumor”) is characterized by a poor response to immunotherapy with monoclonal antibody anti-immune checkpoints.

Another defense mechanism of tumor cells from attack by the immune system is blocking the activity of immune cells by producing immunosuppressive compounds such as prostaglandins, histamine, epinephrine, arginase, TGF-beta and IL-10 [6]. The role of indoleamine dioxygenase (IDO) should be emphasised. It is an enzyme that metabolizes tryptophan, which is necessary for the activation of CTLs and the penetration of T lymphocytes into the tumor. As a result, effector T cells remain at the periphery of the tumor without penetration inside (immune exclusion).

The third mechanism cancer cells can use to escape from immune surveillance is to neutralize immune cells through direct receptor interactions. The CTLA-4 molecule was the first negative immune checkpoint discovered in 1987. The join of the CD80 or CD86 molecules on antigen presenting cells with CTLA-4 on lymphocytes prevents the interaction of CD80/CD86 with the CD28 molecule (a lymphocyte co-stimulatory molecule). The first effective human immunotherapy method was blocking CTLA-4 by anti-CTLA-4 antibody (ipilimumab). However, at present, it seems that blocking other negative immune checkpoints is more important in immunotherapy. The programmed death 1 (PD-1, CD279) is located on the surface of T and B lymphocytes, macrophages and monocytes. It is thought to perform a major role in suppressing the immune system. Under normal conditions, such suppression is beneficial for the body because it protects it from damage during severe inflammatory reaction. Tumor cells develop on their ligands for PD-1: PD-L1 and PD-L2 that bind the PD-1 receptor on T lymphocytes and inhibit their activity [5]. The cancer cells escape from immune surveillance and the cancer cells apoptosis is blocked. These findings enabled the introduction of anti-PD-1 (pembrolizumab and nivolumab) and anti-PD-L1 (atezolizumab and durvalumab) monoclonal antibodies for the treatment of NSCLC patients. In this way, they prevent inactivation of lymphocytes and thus increase their ability to destroy cancer cells (reactivation of T lymphocytes).

Unfortunately, it has been observed that not all NSCLC patients with PD-L1 expression on the surface of tumor cells respond to anti-PD-1 or anti-PD-L1 treatment. The mechanisms that prevent the immune system from recognizing cancer cells and the production of immunosuppressive cytokines, inhibit the effectiveness of treatment. In many patients, it is necessary to use combination therapy, which in the first stage will break the mechanism of avoiding cancer cells’ recognition and stimulate tumor immunogenicity (transition of “cold tumor” into “hot tumor”), and then increase the effectiveness of immune cells in destroying cancer cells [5]. Recent reports indicate that this effect can be achieved through the use of combination therapy involving radiochemotherapy and immunotherapy.

## 2. The Mechanism of Radiation Therapy on Cancer Cells and Its Influence on the Immune System

Radiotherapy is a local treatment method that uses the healing properties of ionizing radiation [7]. Ionizing radiation, acting on the cell, causes the electrons to detach from the orbit of atoms, and thus induces molecular damage in two ways. The first is the shield effect, i.e., direct damage to the most sensitive elements of the cells-DNA, cell membrane and cellular organelles by free electrons. It is responsible for 25% of molecular damages. The second mechanism of action is indirect damage to cell structures by free radicals that were formed during water radiolysis (hydroperoxide radical), which is responsible for 75% of cells damages.

The abscopal effect of radiation therapy is particularly interesting. It was observed that after irradiation, not only the tumor treated with radiation was reduced, but also metastatic lesions distant from this place [8,9,10]. Radiation therapy could have both immunosuppressive effect as well as stimulating effect on the immune system [11,12]. That form of cell death which results in the regulated activation of the immune response is referred to as the immunogenic cell death (ICD). The effect of radiation therapy is cell cycle arrest. At this point, one of the two things may happen: an attempt to repair DNA or induction of apoptosis or necrosis by phagocytosis or autophagy. The damaged tumor cell is fragmented. Apoptotic bodies are formed, which contain tumor antigens. They are recognized and phagocytized by antigen presenting cells (APCs), e.g., dendritic cells that migrate to peripheral lymph nodes and present antigens to T lymphocytes. After the presentation of antigens, T lymphocytes are activated and proliferated and then directed to primary tumor or metastases. Effector T cells have the ability to specifically destroy cancer cells by cytotoxicity. The presence of stimulated cytotoxic T lymphocytes is associated with the abscopal effect of radiation therapy. Stimulated lymphocytes recognize tumor antigens also located in metastatic lesions outside the irradiated areas and they contribute to destruction of metastases. This is of great importance in view of the presence of residual disease, the occurrence of which could be limited by radiation therapy.

Radiation therapy inhibits the repair processes within the DNA of cancer cells and causes mutations in the damaged cell, which results in the formation of neoantigens-structurally changed proteins, specific for tumors, which are identified by the cells of the immune system as foreign and additionally intensify CTLs activation [3].

In addition, dying cancer cells release a number of hallmarks of the immune system, such as a “find me” signal by secreting ATP outside the cell. They also release high mobility group box 1 protein (HMGB1), which acts as a pro-inflammatory cytokine outside the cell. In addition, the expression of calreticulin (CRT) on the surface of the tumor cells is increased [13]. Expression of calreticulin makes cancer cells visible to the immune system. Increased expression of calreticulin on the cell surface sends an “eat me” signal to the cells of the immune system and is a causative agent of phagocytosis and destruction of the cancer cells. Increased concentration of both calreticulin and HMGB1 have been observed in the blood of patients with various cancers. It is thought that high level of calreticulin and HMGB1 is associated with a more advanced tumor process and potentially could act as an unfavorable prognostic marker of disease. However, these factors may have a positive predictive value for immunotherapy used as consolidation treatment after chemoradiotherapy.

Radiotherapy can also activate interferon (IFN) synthesis pathways through activation of stimulator of interferon genes (STING) [14]. Expression of STING mediates the type I interferon production in response to presence of intracellular DNA after DNA fragmentation. STING could sense the presence of intracellular nucleic acids, and then induce interferon β and more than 10 forms of interferon α production. Interferons cause stimulation of cytotoxic T cells and NK cells in a direct way, as well as stimulates expression of PD-L1 molecule on cancer cells.

In summary, radiotherapy destabilizes the function of cancer cells, contributes to the release of tumor antigens and the formation of neoantigens, and affects the production of cytokines, chemokines and other substances that stimulate the activity of immune cells. As a result, low-immunogenic (“cold”) tumors are transformed into highly immunogenic (“hot”, “inflammatory”) tumors, abundant in activated cells of specific immune response (Figure 1) [7,8,12]. Thanks to this, the mechanism of escape of cancer cells from immune surveillance is overcome. Tumor cells are more sensitive to the action of immunotherapy targeted immune checkpoints. The use of anti-PD-1 or anti-PD-L1 antibodies after radiation therapy may contribute to increasing the effectiveness of immunotherapy in patients with non-small cell lung cancer and to prolonging overall survival (OS) and progression free survival (PFS) [15].

## 3. Practical Aspects of the Use of Combination Therapy with Radiotherapy, Chemotherapy and Immunotherapy in NSCLC Patients

Based on the above assumptions, clinical trials were started to prove the effectiveness of combination therapy with immunotherapy and radiotherapy as well as chemotherapy. The KEYNOTE-001 study enrolled patients with advanced non-small cell lung cancer treated in the second and subsequent lines with pembrolizumab in monotherapy [16]. Some patients underwent palliative radiotherapy prior to pembrolizumab therapy (42 patients). 55 patients did not receive radiation therapy. An increase in the median progression-free survival from 2.1 months to 4.4 months and an increase in median overall survival from 5.3 months to 10.7 months have been observed in patients with previous radiotherapy compared to patients without prior radiation therapy. The risk of death was reduced by 42% (HR = 0.58).

In 2020, a multicenter first phase clinical trial was conducted in which pembrolizumab was used concurrently with definitive chemoradiotherapy in NSCLC patients. In this small group of 21 patients, it was shown that combined treatment with pembrolizumab and chemoradiotherapy is tolerable. 69.7% of patients survived without disease progression for over 12 months. However, the results of this study had to be confirmed in a phase 3 clinical trial as performed in the PACIFIC study [17].

In the DETERRED study, concurrent chemoradiotherapy was used in patients with locally advanced NSCLC [18,19]. In first group, chemoradiotherapy followed by atezolizumab consolidation therapy was used (Group 1). In the second group of patients, atezolizumab was added simultaneously to chemoradiotherapy, and then the consolidation of atezolizumab therapy was continued (Group 2). The median OS was 20.1 months in Group 1 and it was not reached in Group 2, but the one-year overall survival rate was 60% in Group 1 and 70% in Group 2. Due to the low numbers of patients (10 and 30 patients), these results were highly unreliable.

Another study, demonstrating the efficacy of combination therapy, was the PACIFIC phase 3 clinical trial, which compared the effectiveness of consolidation therapy with durvalumab (10 mg/kg) vs. placebo in ratio 2:1 every second weeks in patients with stage III inoperable NSCLC without progression after concurrent chemoradiotherapy. These patients had received two or more cycles of platinum-based chemotherapy (containing etoposide, vinblastine, vinorelbine, paclitaxel, docetaxel or pemetrexed) concurrently with definitive radiation therapy at a dose of 54 to 66 Gy [20,21,22,23]. Durvalumab is a human IgG1 monoclonal antibody that exhibits high affinity for the PD-L1 checkpoint and selectively blocks it. PACIFIC trial enrolled 713 patients. In the first observation, the median PFS in the durvalumab group was significantly greater than in the placebo group (16.8 months versus 5.6 months). In 2018, the improvement in median PFS has been maintained (17.2 vs. 5.6 months; HR 0.51, 95% CI [0.41; 0.63]) [20,21]. The 3-year survival rate was 57% vs. 44% in the groups receiving and not receiving durvalumab respectively. There was a 31% decrease in the risk of death (HR = 0.69). The median OS was not reached in durvalumab treated patients but was 29.1 months in patients received placebo. Moreover, the study showed good tolerance of durvalumab. The grade 3 and 4 toxicity rates where 30.5% in the durvalumab arm vs. 26.1% in the placebo arm. Long-term safety was confirmed compared to placebo [20]. On this basis, durvalumab has been registered in consolidation therapy after effective concurrent chemoradiotherapy in patients with locally advanced NSCLC by the U.S. Food and Drug Administration in 2018 and by the European Medicines Agency (EMA) in the European Union countries. To date, it is the only immunotherapy that can be used in this indication.

In the PACIFIC study, all patients received concurrent chemoradiotherapy, which is a more aggressive treatment with potentially more side effects. In some patients it would be advisable to use sequential chemoradiotherapy. Another topic is the validity of using immunotherapy in patients with a mutations in the *EGFR* or *BRAF* genes, *ALK* or *ROS1* genes rearrangements, and low PD-L1 expression [24]. Durvalumab has been approved for the treatment of stage III NSCLC regardless of tumor PD-L1 status and the presence of *EGFR* mutations although the data from the PACIFIC study showed no improvement in survival with durvalumab in patients without PD-L1 expression on tumor cells (HR = 1.14, 95% CI: 0.7–1.84). In the study group, *EGFR* mutation was found in 43 patients (6%), and 74 (10%) patients were PD-L1 negative, therefore the data is insufficient and should be interpreted carefully.

The results of the LUN 14–179 study are worthy of mention. Ninety three (93) patients received concurrent chemoradiation based on cisplatin and pemetrexed, cisplatin and etoposide or carboplatin and paclitaxel (dose of radiation was 59.4 to 66.6 Gy) [25]. Patients who exhibited regression or stabilization of the disease underwent immunotherapy with pembrolizumab. The observation period was 32.2 months. The median target-mediated drug disposition (TMDD) was 30.7 months. Compared to chemoradiotherapy alone, an increase in PFS and OS was achieved in patients treated with combination therapy broken down as follows: 18.7 months (95% CI: 12.4–33.8 months) and 35.8 months (95% CI: 24.2 months-not reached). The incidence of grade 3–5 pneumonitis and other serious adverse events was similar in the group treated with chemoradiotherapy alone and chemoradiotherapy with pembrolizumab consolidation therapy.

Up to now, there have been many valuable reviews analyzing clinical trials with the use of combination therapies based on radiotherapy, chemotherapy and immunotherapy [26,27,28,29]. Some ongoing clinical trials have been completed and current results are available.

The NICOLAS study was the first completed single-armed phase 2 clinical trial with the participation of 79 stage III NSCLC patients [30]. The study evaluated the efficacy of nivolumab (360 mg every 3 weeks) along with platinum-based chemotherapy (3 cycles) with concurrent radiotherapy (66 Gy, 33 fractions). Nivolumab was continued on its own for up to 1 year (480 mg, 4 weeks). The 1-year PFS was 53.7% (95% confidence interval [CI]: 42.0%–64.0%) and the median PFS was 12.7 months (95% CI: 10.1–22.8 months). 37 deaths in the first post-treatment year were observed therefore the 1-year PFS rate at least 45% could not be rejected.

The phase 2 randomized PEMBRO-RT cinical trial ended. The study investigated the effectiveness of combination therapy with pembrolizumab (200 mg/kg every 3 weeks) either alone (control arm) or after stereotactic body radiotherapy (3 doses of 8 Gy) to a single tumor site (experimental arm) in 76 patients with advanced NCSLC. The ORR at 12 weeks was 18% in the control arm vs. 36% in the experimental arm. Median PFS was 1.9 months (95% CI, 1.7–6.9 months) in control arm vs. 6.6 months (95% CI, 4.0–14.6 months) in experimental arm (hazard ratio, 0.71; 95% CI, 0.42–1.18; *p* = 0.19). Median OS was 7.6 months (95% CI, 6.0–13.9 months) vs. 15.9 months (95% CI, 7.1 months to not reached) (hazard ratio, 0.66; 95% CI, 0.37–1.18; *p* = 0.16) [31].

An update of the phase 1 RADVAX clinical trial is also available (Clinical Trial Registration: NCT02303990). The using pembrolizumab and hypofractionated radiotherapy (HFRT) for 24 patients with advanced or metastatic tumor resulted that one patient experienced a complete response, two patients had prolonged responses (9.2 and 28 1 month) and another two experienced prolonged stable disease (7.4 and 7.0 months) [32]. In contrast in 20 patients (83%) grade 1 or 2 treatment-related adverse events were reported.

The current reports of the study are conducted in The University of Texas MD Anderson Cancer Center (Clinical Trial Registration: NCT02239900), which used ipilimumab (3 mg/kg every 3 weeks for 4 doses) with stereotactic ablative radiotherapy (SABR) concurrently 50 Gy (1 day after the first dose) or sequentially 50 Gy or 60 Gy (1 week after the second dose) [33]. In a group of 35 patients, as many as 31 patients responded outside the radiation field. Seven patients (23%) achieved disease stabilization or partial response over 6 months. Furthermore, an increase in the number of peripheral CD8+ T-cells, the CD8+/CD4+ T-cell ratio and 4-1BB and PD1-expressing CD8+ T-cells after irradiation was also demonstrated.

The FORCE phase 2 study is completed, in which 41 patients with advanced NSCLC received nivolumab 240 mg with palliative radiotherapy 5 × 4 Gy (group A) and 60 patients without an indication for radiotherapy received nivolumab only (group B) [34]. ORR was 8.3% in group A while in group B was 23.8%. The low ORR was associated with the need for clinical indications for palliative radiotherapy, which translated into the accumulation of unfavorable features in group A. The clinical trials NCT03224871, NCT02934503 have also ended but the results are still not available

Currently, numerous phase 1. and 2. clinical trials are underway, in which various chemioradiotherapy regimens and radiotherapy methods (usually stereotactic radiotherapy) are used in combination with anti-PD-1 (nivolumab, pembrolizumab) or anti-PD-L1 (atezolizumab, durvalumab, avelumab) antibodies in early stage NSCLC patients (especially if surgery is not available) or locally advanced NSCLC patients [11]. In these studies, radiotherapy or chemoradiotherapy in relation to immunotherapy is used in a sequential, concurrent and inductive manner [11].

In addition, there are many ongoing and future studies using immunotherapy, chemotherapy and radiotherapy in various cinfigurations at all stages of NSCLC (Table 1). The PACIFIC study is continued with PACIFIC-7 and PACIFIC-8 trials that are currently under discussion and preparation. The group of patients that could be enrolled in the study is estimated.

Going deeper, it is worth considering combination immunotherapy based on durvalumab and novel agents. In 2018, a randomized, three-armed clinical trial phase 2 COAST study began (NCT03822351), in which patients without progression after concurrent chemoradiotherapy take part [35,36]. Patients (189) were divided into three groups with consolidation immunotherapy: control arm with durvalumab monotherapy, arm A with durvalumab in combination with oleclumab (human monoclonal antibody targeting the ectonucleotidase CD73), and arm B with durvalumab in combination with monalizumab (immune checkpoint inhibitor targeting NKG2A receptors expressed on tumor cells which prevents inhibition of CD8 + T cells and NK cell). The end of the study is planned on July 2023.

Moreover, chemoradiotherapy could be used in a concurrent or sequential manner [11]. An example would be continuation of the PACIFIC-5 studies, in which durvalumab is used simultaneously with concurrent chemoradiotherapy or after completion of sequential chemoradiotherapy (chemotherapy followed by radiotherapy, at the end of which the consolidation immunotherapy begins) (Figure 2) [11].

NRG-LU004 (NCT03801902) is a clinical trial which investigates impact of accelerated hypofractionated and conventionally fractionated radiotherapy in combination therapy with durvalumab in patients with non-small cell lung cancer in stages II and III [37,38]. Patients received durvalumab 2 weeks before radiation. After immunotherapy in the I arm, patients udergo accelerated hypofractionated radiation therapy (ACRT) 1 fraction per day, 5 days per week for 15 fractions of 60 Gy. In the II arm, patients undergo conventionally fractionated radiation therapy, 1 fraction per day, 5 days per week for 30 fractions of 60 Gy. Durvalumab is going to be continued for 13 cycles.

The selective personalized radioimmunotherapy for locally advanced NSCLC is the subject of SPRINT clinical trial which opened in August 2018 [39]. The essence of the study is to prove safety and better tolerance of combined therapy consisted with durvalumab and radiation therapy which would give a premise to resign from chemotheraphy. Patients with PD-L1 expression on at least 50% of tumor cells will receive 3 cycles of pembrolizumab in dose of 200 mg every second week. Afterwards patient will receive 20 fractions of dose-painted radiotherapy in dose of 55 Gy if lesions with metabolic tumor volume exceeding 20 cm on FDG-PET examination, or 48 Gy while smaller lesions exist. After radiation, pembrolizumab is continued for 12 months. For comparison patients with PD-L1 expression on >50% tumor cells will receive standard chemoradiotherapy or adjuvant therapy.

The above studies could answer the questions about the most optimal method of combination treatment and about a dose and fractionation of radiation therapy. It must not be forgotten that the different doses of radiation and the different methods of radiotherapy fractionation influence the toxicity of these treatment methods. Ladbury et al. ascertained that higher doses of radiation to the immune system were associated with tumor progression and death after the definitive treatment of stage III NSCLC patients. Authors conclude that tailoring radiation therapy to spare the immune system may be an important future direction to improve outcomes in this population of patients. Moreover, NRG Oncology clinical trial RTOG 0617 showed that intensity-modulated radiation therapy (IMRT) was associated with lower rates of severe pneumonitis and cardiac toxicities, which supports routine use of IMRT for locally advanced NSCLC patients [40,41]. Therefore, it should be recognized that the search for the optimal radiation dose to enhance the effect of immunotherapy has significant limitations.

The impact of varied combination of treatment methods on side effects remains also significant. Safety is the vital issue of every treatment. In extensive summary of 13 clinical trials where combination therapy with ICI and CRT was used, presented data says that the frequency of occurrence of grade ≥3 pneumonitis was significantly higher in patients who received anti-PD-1 therapy in comparison to patients who received anti-PD-L1 therapy (8.6%; 95% CI: 6.2%–11.9% vs. 4.4%, 95% CI: 3.0%–6.6%, OR = 2.0; *p* = 0.01). In clinical trials where ICI with CRT were used concurrently, the higher rate of pneumonitis grade 2 was reported in comparison with sequential administration of these methods of treatment (*p* = 0.02) [24,42,43,44].

## 4. Problems and Unknowns of the Combined Use of Radiotherapy, Chemotherapy and Immunotherapy

There is an ongoing discussion about the effectiveness of immunotherapy depending on the number of fractions and the fractional dose of radiation therapy. There is currently no clear data on differences in the effectiveness of individual treatment regimens.

In experiments on mice, the effects on immune system of accelerated hyperfractionated radiotherapy (AHFRT) were compared with conventional fractionated radiotherapy (CFRT) and with hypofractionated radiation therapy using stereotactic ablative radiotherapy (SABR). It has been observed that ablation hypofractionated radiation therapy results in greater stimulation of the immune system, increased tumor infiltration intensity by lymphocytes, reduced recruitment of suppressive myeloid cells within the tumor, and inhibition of vascular endothelial growth factor (VEGF) and its receptor (VEGFR) signaling compared to CFRT [9]. In the next stage of the experiment, mice were subjected to immunotherapy using anti-PD-L1 antibodies. Mice treated with AHFRT combined with anti-PD-L1 antibody showed significantly greater efficacy in controlling tumor growth and extending animal life. Furthermore, it has been observed that AHFRT, by activating the cytotoxic response, is able to cause an abscopal effect and significantly delayed the growth of metastatic tumors outside the irradiation field [9].

Designation of the optimal dose of radiotherapy fractions as well as the number of radiation fractions is a difficult problem to solve [10]. In experiments in mice, it was observed that the dose of 8 Gy ionizing radiation per tumor area in combination with the administration of anti-CTLA-4 antibody resulted in disease remission, tumor size reduction, as well as an abscopal effect on the metastases located outside the radiation area, which contributed to the cure of mice and inhibited the development of residual disease. In another experiment, a 20 Gy dose of radiation was used in combination with an anti-CTLA-4 antibody, which resulted in tumor reduction at the irradiation site, but no abscopal effect on metastatic lesions was observed [15]. After a deeper analysis of this phenomenon, it was found that a high dose of radiation induces the expression of 3′ repair exonuclease 1 (TREX1)-an enzyme that catalyzes DNA degradation. Under its influence, the damaged DNA fragments are being repaired. Cancer cells do not undergo apoptosis, which limits the release of neoantigens. Cancer cells lose immunogenicity, which is a factor that inhibits the stimulation of the immune response and negatively affects the effectiveness of immunotherapy. This experiment showed that the fractional dose of radiotherapy fractions could have an impact on the induction of the immune response and the effectiveness of immunotherapy [15].

The radiation fractionation scheme has also great impact on the activity of the immune system within the tumor. Normally fractionated radiation therapy with a fraction of 1.8–3 Gy induces low PD-L1 expression on tumor cells, weak infiltration of tumor tissue by cytotoxic T lymphocytes, but high infiltration by myeloid suppressive cells, M2 macrophages and T regulatory (Treg) lymphocytes. This situation is not conducive to the effectiveness of immunotherapy used in combination with radiation therapy. A higher fractional dose of 3–10 Gy and hypofractionation induce tumor tissue infiltration by cytotoxic T lymphocytes with high expression of the T-cell immunoreceptor with Ig and ITIM domains (TIGIT, polyovirus receptor, PVR, CD155), high expression of PD-L1 on cancer cells, but also intensive infiltration of tumor tissue by Treg lymphocytes. This situation prompts the use of anti-PD-1 or anti-PD-L1 and anti-TIGIT immunotherapy (in clinical studies), as well as cyclophosphamide, which reduces the activity of Treg lymphocytes. A very high dose of radiation above 10 Gy together with its hypofractionation causes a strong infiltration of tumor tissue by T cells expressing PD-1 and TIGIT molecules, vascular damage and serious side effects [21].

In the search for answers to the above problems, questions about the selection of the immunotherapy method depending on the fractionation method and dose of radiation therapy arise. Individual radiotherapy or chemoradiotherapy regimens produce specific molecular and immune effects in the tumor tissue as well as throughout the body. These changes may directly translate into increased or decreased effectiveness of selected immunotherapy methods.

## 5. Conclusions

The use of radiotherapy or radiochemotherapy in combination with immunotherapy seems to be a good direction for the development of new therapeutic strategies in patients with non-small cell lung cancer. The discovery of the impact of ionizing radiation on the stimulation of the immune system initiated numerous clinical trials. The first reports of the results of these studies (PACIFIC trial results and durvalumab registration) are very promising. However, there are still many unknowns regarding the planning of combination therapies. First, it is necessary to identify a group of patients in which such treatment will be effective. We currently do not know the precise predictors in the qualification for immuno(chemo)radiotherapy. The possibility of qualification of such treatments is decided by the patients’ performance status or stage of the disease, but the predictive factors for the combination therapy with immunotherapy, such as PD-L1 expression on tumor cells or tumor mutation burden (TMB) have not been sufficiently studied. The next task is to determine the optimal combination regimen. Small changes in individual strategies of treatment could bring significant differences in its effectiveness. It is necessary to determine the appropriate sequence of combination therapy, as well as the number of fractions, the fractional dose and total dose of radiation therapy. Depending on the choice of irradiation method, we should choose the method of immunotherapy and specify its duration. Another problem may be new side effects of such treatments that are difficult to predict. Undoubtedly, previous discoveries are optimistic. The completion of ongoing clinical trials will certainly bring many answers and will be a valuable source of knowledge for scientists and further discoveries in this field.

## Figures and Tables

**Figure 1 cancers-13-01222-f001:**
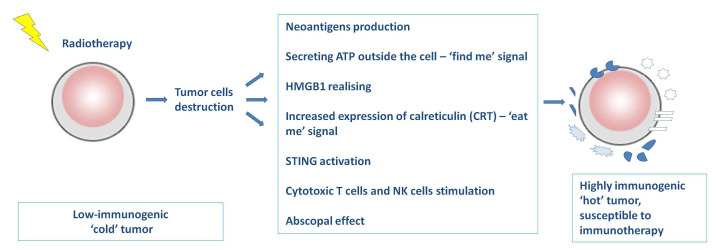
The mechanism of radiation therapy on cancer cells and its influence on the immune system.

**Figure 2 cancers-13-01222-f002:**
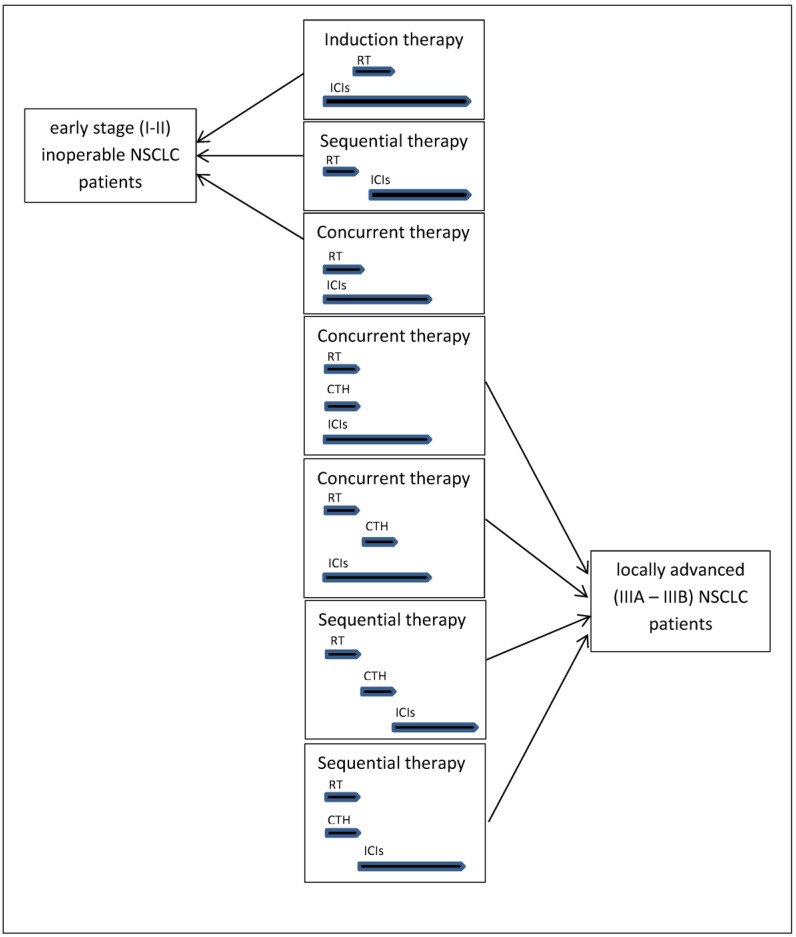
Different combination treatment regimens containing radiotherapy or radiochemotherapy and immunotherapy in patients in various stages of NSCLC.

**Table 1 cancers-13-01222-t001:** Ongoing and future clinical trials investigating combined treatment based on immunotherapy and radiotherapy or radiochemotherapy in NSCLC patients.

Clinical Trial Identifier	Treatment Method	Stage of NSCLC	Phase	Estimated Enrollment	Status
NCT04245514 (SAKK 16/18)	Durvalumab + RT (3 cohorts)	III	2	90	Recruiting
NCT04202809 (ESPADURVA)	Chemo- and radiochemotherapy ± Durvalumab	IIIA–IIIB	2	90	Recruiting
(PACIFIC-7)	Durvalumab + radiochemotherapy flollowed by durvalumab + tremelimumab	III	3	n/a	Not yet recruiting
(PACIFIC-8)	Domvanalimab + platinum-based radiochemotherapy	III	3	n/a	Not yet recruiting
NCT03706690 (PACIFIC-5)	Durvalumab vs. placebo	III	3	360	Recruiting
NCT03801902 (ARCHON-1)	Durvalumab + ACRT vs. durvalumab + CFRT	II–III	1	24	Recruiting
NCT03523702 (SPRINT)	Accelerated, dose-painted RT + pembrolizumab vs. accelerated dose-painted RT + chemotherapy (carboplatin + paclitaxel)	II–III	2	63	Recruiting
NCT03176173(RRADICAL)	Nivolumab, pembrolizumab, atezolizumab ± RT	IV	2	85	Recruiting
NCT04776447 (APOLO)	Atezolizumab + chemotherapy (carboplatin + paclitaxel) + RT	IIIA–IIIB	2	51	Not yet recruiting
NCT02839265 (FLT3)	FLT3 Ligand Therapy (CDX-301) + SBRT	III–IV	2	29	Active, not recruiting
NCT03383302 (STILE)	Nivolumab + SBRT	I–II	1/2	31	Recruiting
NCT03965468 (CHESS)	Durvalumab + chemotherapy (carboplatin + paclitaxel) + SBRT	IV	2	47	Recruiting
NCT03825510 (I-SABR)	SBRT + nivolumab vs. SBRT + pembrolizumab	IV	n/a	100	Recruiting
NCT03644823 (COM-IT-1)	Atezolizumab + low dosed RT	III–IV	2	30	Recruiting
NCT03110978 (I-SABR)	SBRT ± nivolumab	I–IIa	2	140	Recruiting
NCT04245514	Durvalumab + RT (3 cohorts)	III	2	90	Recruiting
NCT03774732 (NIRVANA-LUNG)	Pembrolizumab + paclitaxel ± 3D-CRT/SABR	III–IV	3	460	Recruiting
NCT03867175	Pembrolizumab ± SBRT	IV	3	112	Recruiting
NCT03168464 (BMS # CA209-632)	Nivolumab + ipilimumab + non-ablative RT	IV	1/2	45	Recruiting
NCT04230408 (PACIFIC BRAZIL)	Durvalumab + chemotherapy (carboplatin + paclitaxel) + RT	III	2	48	Recruiting
NCT03223155 (COSINR)	SBRT + nivolumab/ipilimumab	III	1	80	Recruiting
NCT04577638 (AIRING)	Nivolumab + IMRT	III	2	60	Not yet recruiting
NCT04372927 (ADMIRAL)	Chemotherapy (cisplatin + etopozyd/cisplatin + pemetrexed) + durvalumab + RT	III	2	40	Not yet recruiting
NCT04765709 (BRIDGE)	Chemotherapy (cisplatin/carboplatin + vinorelbine/pemetrexed) + durvalumab + RT	III	2	65	Not yet recriuiting
NCT03916419	Chemotherapy (cisplatin + paclitaxel) + RT + durvalumab	IIB–IIIA	2	27	Not yet recruiting
NCT03275597	Dual checkpoint inhibition (durvalumab + tremelimumab) + SBRT	IV	1b	31	Recruiting
NCT03237377	Durvalumab + RT vs. durvalumab + tremelimumab + RT	III	2	32	Recruiting
NCT04654520	Chemotherapy + IMRT ± immunotherapy	IV	n/a	290	Not yet recruiting
NCT04151940	Chemoimmunotherapy + RT	IV	n/a	40	Recruiting
NCT03808337 (PROMISE-005)	Systemic Therapy/Standard of Care + SBRT	IV	2	142	Recruiting
NCT03391869 (LONESTAR)	Nivolumab + ipilimumab vs. nivolumab + ipilimumab + RT	IV	3	270	Recruiting
NCT02444741	Pemnrolizumab + SBRT vs. pembrolizumab + IMRT/PBRT/3D-CRT vs. pembrolizumab + RT upon PD	IV	1/2	124	Active, not recruiting
NCT04310020	SBRT + atezolizumab	II–III	2	47	Recriuting
NCT03871153	Neoadjuvant chemotherapy (carboplatin + paclitaxel) + RT + durvalumab	III	2	25	Recruiting
NCT03050060 (ImmunoRad)	Nelfinavir + pembrolizumab/atezolizumab/nivolumab + RT	IV	2	120	Recruiting
NCT02888743	Durvalumab + tremelimumab vs. durvalumab + tremelimumab + RT	IV	2	180	Active, not recruiting
NCT04214262	SBRT ± atezolizumab	I–II	3	480	Recruiting
NCT04650490	Immunotherapy + SRS	IV	2	80	Not yet recruiting
NCT04271384	Nivolumab + SABR	I	2	30	Recruiting
NCT03337698 (Morpheus-Lung)	Multiple immunotherapy-based treatment combinations ± RT	IV	1/2	380	Recruiting
NCT03446547 (ASTEROID)	SBRT ± durvalumab	I	2	216	Recruiting
NCT03141359	SBRT + chemotherapy (cisplatin +etoposide/carboplatin + paclitaxel) ± durvalumab	II–III	2	60	Recruiting
NCT04597671 (NVALT 28/ PRL01)	Durvalumab ± low-dose PCI	IV	3	170	Not yet recruiting
NCT04092283	Chemotherapy (cisplatin + etoposide/pemetrexed/carboplatin + paclitaxel) + RT +durvalumab	III	3	660	Recruiting
NCT04291092 (SHR-1210)	Immunotherapy + SRS	IV	2	20	Not yet recruiting
NCT03158883	Avelumab + SABR	IV	Early phase 1	26	Recruiting
NCT03915678 (AGADIR)	Atezolizumab + BDB001 + RT	IV	2	247	Not yet recruiting
NCT03509012 (CLOVER)	Durvalumb + cisplatin + etoposide chemotherapy + RT vs. durvalumab + carboplatin + paclitaxel chemotherapy + RT vs. chemotherapy only	III	1	105	Active, not recruiting
NCT04167657 (STAR)	Sintilimab + RT	IIIB–IV	2	37	Recruiting
NCT04540757 (PIONEER)	RT + immunotherapy/chemotherapy ± surgery	III	n/a	66	Recruiting
NCT04434560	SRS ± nivolumab/ipilimumab	IV	2	40	Recruiting
NCT03102242	Atezolizumab + carboplatin + paclitaxel chemotherapy + RT	IIIA–IIIB	2	64	Active, not recruiting
NCT04023812 (MOOREA)	chemotherapy, targeted therapy, immunotherapy, anti-angiogenesis therapy and radiotherapy	III	n/a	700	Recruiting

Abbreviations: RT—radiotherapy, SRS—stereotactic radiosurgery, 3D-CRT—three-dimensional conformal radiation therapy, SBRT—stereotactic body radiation therapy, SABR—stereotactic ablative radiation therapy, IMRT—intensity-modulated radiation therapy, ACRT—accelerated hypofractionated radiation therapy, CFTR—conventionally fractionated radiation therapy, PCI—prophylactic cranial irradiation, n/a—not available.

## Data Availability

Not applicable.

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
