# Peer review of "Molecular and Clinical Premises for the Combination Therapy Consisting of Radiochemotherapy and Immunotherapy in Non-Small Cell Lung Cancer Patients"

_cancers, 2021, doi:10.3390/cancers13061222_

Round 1

Reviewer 1 Report

The presented manuscript entitled "Molecular and clinical premises for the combination therapy consisting of radiochemotherapy and immunotherapy in
non-small cell lung cancer patients" is extremely interesting and presents current observations on the molecular mechanisms in the development of lung cancer and how different types of therapy affect them.

The authors describe the methods of treatment and make comparisons in their various combinations, including in detail the importance on one's own antitumor immunity and the neoplasmic response.
These references are up-to-date.
The language and style are easy to understand.

Author Response

Point 1. The presented manuscript entitled "Molecular and clinical premises for the combination therapy consisting of radiochemotherapy and immunotherapy in
non-small cell lung cancer patients" is extremely interesting and presents current observations on the molecular mechanisms in the development of lung cancer and how different types of therapy affect them.

The authors describe the methods of treatment and make comparisons in their various combinations, including in detail the importance on one's own antitumor immunity and the neoplasmic response.
These references are up-to-date.
The language and style are easy to understand.

Response 1:

We are very grateful for Your comment. Your opinion is very valuable to us and contributes to even greater diligence in our work.

Reviewer 2 Report

I read the manuscript of Frak et al. with a pleasure. This is a carefully written review article that is worthy of publication, as it concerns one of the most important and still-evolving field of cancer therapy. However, given that this topic have also been nicely reviewed by others [*], I wound be important to highlight (in their response to the review) the novelty of the present article, i.e what new aspects it discussed over the previously published?

*Lazzari et al. 2018: Combination of immunotherapy with chemotherapy and radiotherapy in lung cancer: is this the beginning of the end for cancer?

*Ko et al. 2018: The Integration of Radiotherapy with Immunotherapy for the Treatment of Non-Small Cell Lung Cancer.

*Bhalla et al. 2018: Combining immunotherapy and radiotherapy in lung cancer.

*Spaas et al. 2019: Is the Combination of Immunotherapy and Radiotherapy in Non-small Cell Lung Cancer a Feasible and Effective Approach?

Minor comments:

line 101: The join of the CD80 or CD 86

line 131 and 162: “The first is the shield effect, i.e. direct damage to the most sensitive elements of the cells - DNA, cells’ membrane and cells’ organelles by a free electrons. It is responsible for 25% of molecular damages. The second mechanism of action is indirect damage to cells’ structures by free. …. on the cells’ surface” Please consider ‘cell membrane’, ‘cellular organelles’, ‘cell structures’ or ‘cellular structures’, ‘cell surface’ .

Line 139: Radiation therapy could have both  immunosuppressive effect as well as ‘stimulating effect in the immune system stimulating effect in the immune system [11,12]. Please consider ‘stimulating effect on the immune system’

Line 140: “The effect of radiation therapy is cell cycle arrest and attempt to repair DNA or induction of apoptosis or necrosis by phagocytosis or autophagy.” Please rephrase, as the underlined part of the sentence is obscure.

e.g. “HMGB1 (high mobility group box 1 protein)”, then “calreticulin (CRT)”, then “Radiotherapy can also activate interferon (IFN) synthesis pathways through activation of stimulator of interferon genes (STING)”

Please uniform - e.g. at first appearance in the text  - the full name followed by abbreviation in brackets.

In the section 2 on radiation therapy, please consider introducing the term ‘immunogenic cell death (ICD)’

Line 308: “Authors conclude tah tailoring radiation therapy”

Author Response

At the begining we would like to thank You for Your favourable comment and objections. It is very valuable to us and contribute to the amendment of our work and it teach us how to improve our work in the future.

Point 1: I read the manuscript of Frak et al. with a pleasure. This is a carefully written review article that is worthy of publication, as it concerns one of the most important and still-evolving field of cancer therapy. However, given that this topic have also been nicely reviewed by others [*], I wound be important to highlight (in their response to the review) the novelty of the present article, i.e what new aspects it discussed over the previously published?

*Lazzari et al. 2018: Combination of immunotherapy with chemotherapy and radiotherapy in lung cancer: is this the beginning of the end for cancer?

*Ko et al. 2018: The Integration of Radiotherapy with Immunotherapy for the Treatment of Non-Small Cell Lung Cancer.

*Bhalla et al. 2018: Combining immunotherapy and radiotherapy in lung cancer.

*Spaas et al. 2019: Is the Combination of Immunotherapy and Radiotherapy in Non-small Cell Lung Cancer a Feasible and Effective Approach?

Response 1:  The mentioned articles are very carefully prepared materials describing the topic of combination therapies with the use of radiotherapy, chemotherapy and immunotherapy. I supplemented the thesis with the results of clinical trials, which were still in progress in the articles presented above. In this way, I updated the available knowledge and introduced new information from recently completed research.  I added following text in the section 3, in paragraph 7. 

Up to now, there have been many valuable reviews analyzing clinical trials with the use of combination therapies based on radiotherapy, chemotherapy and immunotherapy [26,27,28,29]. Some ongoing clinical trials have been completed and current results are available.

The NICOLAS study was the first completed single-armed phase 2 clinical trial with the participation of 79 stage III NSCLC patients [30]. The study  evaluated the efficacy of nivolumab (360 mg every 3 weeks) along with platinum-based chemotherapy (3 cycles) with concurrent radiotherapy (66 Gy, 33 fractions). Nivolumab was continued on its own for up to 1 year (480 mg, 4 weeks). The 1-year PFS was 53.7% (95% confidence interval [CI]: 42.0% –64.0%) and the median PFS was 12.7 months (95% CI: 10.1–22.8 months). 37 deaths in the first post-treatment year were observed therefore the 1-year PFS rate at least 45% could not be rejected.

The phase 2 randomized PEMBRO-RT cinical trial ended. The study investigated the effectiveness of combination therapy with pembrolizumab (200 mg/kg every 3 weeks) either alone (control arm) or after stereotactic body radiotherapy (3 doses of 8 Gy) to a single tumor site (experimental arm) in 76 patients with advanced NCSLC. The ORR at 12 weeks was 18% in the control arm vs 36% in the experimental arm. Median PFS was 1.9 months (95% CI, 1.7-6.9 months) in control arm vs 6.6 months (95% CI, 4.0-14.6 months) in experimental arm (hazard ratio, 0.71; 95% CI, 0.42-1.18; P = .19). Median OS was 7.6 months (95% CI, 6.0-13.9 months) vs 15.9 months (95% CI, 7.1 months to not reached) (hazard ratio, 0.66; 95% CI, 0.37-1.18; P = .16)[31].

An update of the phase 1 RADVAX clinical trial is also available (Clinical Trial Registration: NCT02303990). The using pembrolizumab and hypofractionated radiotherapy (HFRT) for 24 patients with advanced or metastatic tumor resulted that one patient experienced a complete response, two patients had prolonged responses (9.2 and 28 1 month) and another two experienced prolonged stable disease (7.4 and 7.0 months) [32]. In contrast in 20 patients (83%) grade 1 or 2 treatment-related adverse events were reported.

The are current reports of the study conducted in The University of Texas MD Anderson Cancer Center (Clinical Trial Registration: NCT02239900), which used ipilimumab (3 mg/kg every 3 weeks for 4 doses) with stereotactic ablative radiotherapy (SABR) concurrently 50 Gy (1 day after the first dose) or sequentially 50 Gy or 60 Gy (1 week after the second dose) [33]. In a group of 35 patients, as many as 31 patients responded outside the radiation field. Seven patients (23%) achieved disease stabilization or partial response over 6 months. Furthermore, an increase in the number of peripheral CD8+ T-cells, the CD8+/CD4+ T-cell ratio and 4-1BB and PD1-expressing CD8+ T-cells after irradiation was also demonstrated.

The FORCE phase 2 study is completed, in which 41 patients with advanced NSCLC received nivolumab 240 mg with palliative radiotherapy 5 x 4 Gy (group A) and 60 patients without an indication for radiotherapy received nivolumab only (group B) [34]. ORR was 8.3% in group A while in group B was 23.8%. The low ORR was associated with the need for clinical indications for palliative radiotherapy, which translated into the accumulation of unfavorable features in group A.

 There are also ended clinical trials NCT03224871, NCT02934503 but results are still not available

According to the text I actualized references in points 26-34:

  1. Lazzari C, Karachaliou N, Bulotta A, Viganó M, Mirabileet A et al. Combination of immunotherapy with chemotherapy and radiotherapy in lung cancer: is this the beginning of the end for cancer?. Ther Adv Med Oncol. 2018;10:1758835918762094
  2. Ko EC, Raben D, Formenti SC. The Integration of Radiotherapy with Immunotherapy for the Treatment of Non–Small Cell Lung Cancer. Clin Cancer Res 2018;(24)(23): 5792-5806
  3. Bhalla N, Brooker R, Brada M. Combining immunotherapy and radiotherapy in lung cancer. J Thorac Dis. 2018;10(Suppl 13):S1447-S1460.
  4. Spaas M, Lievens Y. Is the Combination of Immunotherapy and Radiotherapy in Non-small Cell Lung Cancer a Feasible and Effective Approach?. Front Med (Lausanne). 2019;6:244.
  5. Peters S, Felip E, Dafni U, Tufman A, Guckenberger M et al. Progression-Free and Overall Survival for Concurrent Nivolumab With Standard Concurrent Chemoradiotherapy in Locally Advanced Stage IIIA-B NSCLC: Results From the European Thoracic Oncology Platform NICOLAS Phase II Trial (European Thoracic Oncology Platform 6-14). Journal of Thoracic Oncology 2021;16(2):278–88.
  6. Theelen WSME, Peulen HMU, Lalezari F, van der Noort V, de Vries JF et al. Effect of Pembrolizumab After Stereotactic Body Radiotherapy vs Pembrolizumab Alone on Tumor Response in Patients With Advanced Non–Small Cell Lung Cancer: Results of the PEMBRO-RT Phase 2 Randomized Clinical Trial. JAMA Oncol. 2019;5(9):1276–1282.
  7. Maity A, Mick R, Huang AC, George SM, Farwell MD et al. A phase I trial of pembrolizumab with hypofractionated radiotherapy in patients with metastatic solid tumours. Br J Cancer 2018;119,1200–1207.
  8. Tang C, Welsh JW, de Groot P, Massarelli E, Chang JY et al. Ipilimumab with Stereotactic Ablative Radiation Therapy: Phase I Results and Immunologic Correlates from Peripheral T Cells. Clin Cancer Res. 2017 Mar 15;23(6):1388-1396.
  9. Bozorgmehr F, Fischer JR, Bischof M, Atmaca A, Wetzel S et al. LBA58 - ORR in patients receiving nivolumab plus radiotherapy in advanced non-small cell lung cancer: First results from the FORCE trial. Annals of Oncology (2020) 31 (suppl_4): S1142-S1215.

Point 2:  line 101: The join of the CD80 or CD 86

Response 2:  I changed to: The join of the CD80 or CD86

Point 3: line 131 and 162: “The first is the shield effect, i.e. direct damage to the most sensitive elements of the cells - DNA, cells’ membrane and cells’ organelles by a free electrons. It is responsible for 25% of molecular damages. The second mechanism of action is indirect damage to cells’ structures by free. …. on the cells’ surface” Please consider ‘cell membrane’, ‘cellular organelles’, ‘cell structures’ or ‘cellular structures’, ‘cell surface’ .

Response 3: I changed to ‘cell membrane’, ‘cellular organelles’, ‘cell structures’, ‘cell surface’.

Point 4: Line 139: Radiation therapy could have both  immunosuppressive effect as well as ‘stimulating effect in the immune system stimulating effect in the immune system [11,12]. Please consider ‘stimulating effect on the immune system’

Response 4: I changed the sentence to ‘Radiation therapy could have both  immunosuppressive effect as well as ‘stimulating effect on the immune system stimulating effect in the immune system [11,12].’ 

Point 5: Line 140: “The effect of radiation therapy is cell cycle arrest and attempt to repair DNA or induction of apoptosis or necrosis by phagocytosis or autophagy.” Please rephrase, as the underlined part of the sentence is obscure.

Response 5: I changed the sentence to: ‘The effect of radiation therapy is cell cycle arrest. At this point, one of the two occurrence may appear: attempt to repair DNA or induction of apoptosis or necrosis by phagocytosis or autophagy.’

Point 6:  e.g. “HMGB1 (high mobility group box 1 protein)”, then “calreticulin (CRT)”, then “Radiotherapy can also activate interferon (IFN) synthesis pathways through activation of stimulator of interferon genes (STING)” Please uniform - e.g. at first appearance in the text  - the full name followed by abbreviation in brackets.

Response 6: I uniformed text, I put full name followed by abbreviation in brakets.

In lines 61, 68-72, 81, 97, 160, 336, 352, 367.

Point 7:  In the section 2 on radiation therapy, please consider introducing the term ‘immunogenic cell death (ICD)’

Response 7: I introduced the term immunogenic cell death (ICD) in lines 140 and 141.

Point 8: Line 308: “Authors conclude tah tailoring radiation therapy”

Response 8: I changed the sentence to “Authors conclude that tailoring radiation therapy”

Reviewer 3 Report

This is an interesting review on the rationale for using radiation treatment in combination with chemotherapy followed by IO in patients with inoperable/advanced NSCLC.

I do not have major comments.

Author Response

Point 1: This is an interesting review on the rationale for using radiation treatment in combination with chemotherapy followed by IO in patients with inoperable/advanced NSCLC.

I do not have major comments.

Response 1:  We are very grateful for Your comment. Your opinion is very valuable to us and contributes to even greater diligence in our work.